# Bioinformatics and Machine Learning Approaches to Understand the Regulation of Mobile Genetic Elements

**DOI:** 10.3390/biology10090896

**Published:** 2021-09-10

**Authors:** Ilektra-Chara Giassa, Panagiotis Alexiou

**Affiliations:** Central European Institute of Technology (CEITEC), Masaryk University, 625 00 Brno, Czech Republic; igiassa@mail.muni.cz

**Keywords:** transposable elements regulation, mobile genetic elements, machine learning, bioinformatics methods, DNA methylation, small RNAs, PIWI-interacting RNAs, circular RNAs

## Abstract

**Simple Summary:**

Transposable elements (TEs) are DNA sequences that are, or were, able to move (transpose) within the genome of a single cell. They were first discovered by Barbara McClintock while working on maize, and they make up a large fraction of the genome. Transpositions can result in mutations and they can alter the genome size. Cells regulate the activity of TEs using a variety of mechanisms, such as chemical modifications of DNA and small RNAs. Machine learning (ML) is an interdisciplinary subject that studies computer algorithms that can improve through experience and by the use of data. ML has been successfully applied to a variety of problems in bioinformatics and has exhibited favorable precision and speed. Here, we provide a systematic and guided review on the ML and bioinformatic methods and tools that are used for the analysis of the regulation of TEs.

**Abstract:**

Transposable elements (TEs, or mobile genetic elements, MGEs) are ubiquitous genetic elements that make up a substantial proportion of the genome of many species. The recent growing interest in understanding the evolution and function of TEs has revealed that TEs play a dual role in genome evolution, development, disease, and drug resistance. Cells regulate TE expression against uncontrolled activity that can lead to developmental defects and disease, using multiple strategies, such as DNA chemical modification, small RNA (sRNA) silencing, chromatin modification, as well as sequence-specific repressors. Advancements in bioinformatics and machine learning approaches are increasingly contributing to the analysis of the regulation mechanisms. A plethora of tools and machine learning approaches have been developed for prediction, annotation, and expression profiling of sRNAs, for methylation analysis of TEs, as well as for genome-wide methylation analysis through bisulfite sequencing data. In this review, we provide a guided overview of the bioinformatic and machine learning state of the art of fields closely associated with TE regulation and function.

## 1. Introduction

Among the several types of genomic repeated sequences in the human genome, transposable elements comprise the largest fraction, estimated to approximately half, or even much larger, TE fractions [1,2]. These TEs, often called transposons or jumping genes, are DNA sequences that have, or once had, the ability to move within the genome, either directly or through an RNA intermediate. TEs contribute to the large variety of the genome size across species that cannot be attributed solely to the number of genes it contains (C-value paradox). TEs are present, to largely varying degrees, in the genomes of all known types of organisms, both prokaryotic and eukaryotic. Some species even show a larger number of genomic transposons than host sequences [3,4,5,6]. Several publications have discussed the numerous and diverse types and roles of TEs [7,8,9,10,11,12,13,14,15,16] and the mechanisms responsible for their regulation [17,18,19,20,21,22,23,24,25,26,27,28,29,30,31,32,33,34,35], as well as the available bioinformatic tools for the analysis of TEs [36]. In short, the top of the classification hierarchy categorizes TEs in two major types. Class I TEs, also called retrotransposons, transpose via an RNA intermediate that is subsequently reverse transcribed to cDNA before inserting elsewhere in the genome. Class II TEs, also called DNA transposons, directly excise themselves from one location before reinsertion. Retrotransposons can be distinguished in long terminal repeat (LTR) elements, which are more abundant in plants, and in non-LTR elements, from which long interspersed nuclear elements (LINEs) enrich most animal genomes. A considerable portion (close to 20%) of the human and mouse genomes are comprised of LINE-1 [37,38]. The vast majority of TEs in the human genome have lost the ability to fully mobilize [39,40,41]. The exception to that are 80–100 human-specific LINE-1 element (L1HS) that are the only fully autonomous TE with the ability to generate new transposition events in human to date [41]. However, most TEs have retained some level of functionality, including the ability to direct their own transcription.

Thus, transcriptome-wide sequencing assays, like RNA-seq, frequently include transposon-derived transcripts among the set of expressed sequences. Moreover, some transposon transcripts have been co-opted to play a role in host function. Particularly during early development, some expressed transposon transcripts have been shown to be necessary for proper cell differentiation and maintenance of identity [42,43,44,45,46]. In addition to their roles in general cellular function, several types of transposons have become intricately entangled within gene regulatory networks, [47] contributing both to cis-regulatory sequences [48,49,50] as well as general chromatin environments [51,52,53]. For this reason, it is paramount that we consider the contribution of repetitive elements as we unravel the genomic and epigenomic landscapes of gene expression regulation.

Even though most TEs affect the host only in a neutral fashion, certain transpositions might have a deleterious effect. These transpositions can disrupt gene function or cause compromising chromosomal rearrangements, as evident in the more than 120 diseases connected to TE insertions in human [54,55,56,57]. Selective pressure on host organisms has driven the development of diverse transcriptional and post-transcriptional mechanisms to suppress and control TE activity. Within those mechanisms fall small RNAs, ATP-dependent chromatin remodelers, DNA methylation, and Krüppel-associated box (KRAB) zinc-finger proteins (ZFPs), the latter thought to have evolved along with TEs [49,58,59,60,61,62].

Machine learning (ML) is the development and application of computer algorithms that are able to improve a performance criterion through experience and by the use of data to build a model [63,64]. ML has been successfully applied to a variety of bioinformatics problems in the fields of biology and medicine [65,66], with favorable results in terms of precision and speed. Recent advancements in experimental techniques have led to the exponential growth of the amount of biological data. That growth resulted in two major issues: the storage and management of big data and their meaningful interpretation. ML has already been extensively used for knowledge discovery via converting that bulk of data into biological knowledge of the underlying mechanisms in the form of testable models [67].

Machine learning (ML) is a multidomain interdisciplinary subject. There are many different statistical, probabilistic, and optimization techniques that can be implemented as the learning methods. Logistic regression, artificial neural networks (ANN), random forest (RF), and Support Vector Machine (SVM) are frequently used learning methods. Artificial neural network (ANN) models are powerful tools in ML. They can approximate functions and dynamics by learning from examples and were originally meant to simulate the functioning of a human brain [68]. A Convolutional Neural Network (CNN) is a type of ANN inspired by biological processes in the sense that the connectivity pattern between neurons resembles the organization of the animal visual cortex [69]. Random forest (RF) is an ensemble method that constructs a multitude of decision trees serving for regression and classification tasks [70]. For classification purposes, an RF can construct multiple independent decision trees from the original features of the training data set and then fuse all trees by voting to obtain an optimal model. Support Vector Machines (SVMs) are supervised learning models that can perform (linear or non-linear) classification and regression analysis. SVMs are very powerful at recognizing subtle patterns in complex datasets [71].

For the assessment of classification tasks, a set of useful metrics are used. Sensitivity or recall (Sn, also called true positive rate, TRP) is the proportion of true positive observations. Specificity (Sp) is the proportion of true negative observations. Accuracy (Acc) is the ratio of true positive observations to the total observations. Precision (Pre) is the ratio of true positive observations to the total predicted positive observations, and Matthew’s Correlation Coefficient (MCC) is a measure of the quality of binary classification. In short, the metrics are defined as such:(1)$$\mathrm{Sn}=\frac{\mathrm{TP}}{\mathrm{TP}+\mathrm{FN}}$$
(2)$$\mathrm{Sp}=\frac{\mathrm{TN}}{\mathrm{TN}+\mathrm{FP}}=1-\mathrm{FPR}$$
(3)$$\mathrm{Acc}=\frac{\mathrm{TN}+\mathrm{TP}}{\mathrm{TN}+\mathrm{TP}+\mathrm{FP}+\mathrm{FN}}$$
(4)$$\mathrm{Pre}=\frac{\mathrm{TP}}{\mathrm{TP}+\mathrm{FP}}$$
(5)$$\mathrm{MCC}=\frac{\mathrm{TP}\times \mathrm{TN}-\;\mathrm{FP}\times \mathrm{FN}}{\sqrt{\left(\mathrm{TP}+\mathrm{FP}\right)\left(\mathrm{TP}+\mathrm{FN}\right)\left(\mathrm{TN}+\mathrm{FP}\right)\left(\mathrm{TN}+\mathrm{FN}\right)}}$$

TP and FP are the numbers of true positive and false positive assessments, respectively. Additionally, TN and FN are the numbers of true negative and false negative assessments, respectively, and FPR is the false positive rate.

Another widely used measure of performance is the Receiver Operator Characteristic (ROC) curve. ROC is a probability plot of the TPR against the FPR for various discrimination thresholds, and practically indicates the diagnostic ability of a binary classifier. A summary of the ROC curve is the area under the curve (AUC). The higher the AUC, the better the performance of the classifier at distinguishing the positive and negative classes. Lastly, an important technique for the validation of ML models is the so-called cross-validation (CV). CV is used to test the ability of the model to predict new data (data not used when training and testing the model). In the case of the *k*-fold CV (abbreviated as *k*-CV, with 10-CV being commonly used), the original dataset is partitioned into *k* subdatasets of equal size. One of the datasets is used as the validation dataset and the remaining *k*-1 are used for the training of the model. This process is repeated *k* times.

Here we present a systematic review of the numerous bioinformatics and ML methods and tools employed in the analysis of the mechanisms that regulate the expression of TEs. We discuss the basic features of each method, potential comparative performance among them, as well as data and code availability. For easier navigation through the review, the various tools are grouped based on the primary analysis they perform: on DNA methylation or small RNA data. In the latter case, the tools that serve for the analysis of a single small RNA type (namely PIWI-interacting RNAs, circular RNAs) are grouped separately.

## 2. Methods and Techniques

### 2.1. Analysis of DNA Methylation

A safeguard mechanism to suppress the activity of TEs is the methylation of cytosine nucleotides to produce 5-methylcytosine (5mC), a modification that can induce transcriptional silencing of the methylated locus. Cytosine DNA methylation is a stable epigenetic mark that is critical for diverse biological processes, such as gene and transposon silencing, imprinting, and X chromosome inactivation [72]. Bisulfite sequencing (BS) is a powerful technique for the study of DNA cytosine methylation [73]. In tandem with next-generation sequencing (NGS) technology, it can potentially detect the methylation status of every cytosine in the genome. Bisulfite treatment of DNA does not affect the 5-methylcytosines; however, non-methylated Cytosines are converted into Uracils, which are further converted into Thymines during the subsequent PCR amplification. This two-step process produces four individual strands of DNA for any given genomic locus, all of which can potentially appear in a sequencing experiment.

There are two distinct types of bisulfite libraries: in the first case (BS-seq), the sequencing library is generated in a directional manner, so that the actual sequencing reads correspond to a bisulfite converted version of either the original forward or reverse strand [74], whereas in the second case (MethylC-seq), the strand specificity is not preserved [75]. In the latter case, all four possible bisulfite DNA strands are sequenced at roughly the same frequency.

The task of mapping the bisulfite-treated sequences to a reference genome is a significant computational challenge due to a number of reasons: reduced complexity of the DNA code, there are up to four DNA strands to be analyzed, and the fact that each read can theoretically exist in all possible methylation states. Ambiguous reads (reads that map to both the converted and unconverted reference genomes) are a great challenge in BS sequencing so that many relevant software packages avoid that issue by not including multimapped reads, among them BSMAP [76], Bismark [77], MOABS [78], and BS-Seeker3 [79]. On the other hand, tools like TEPID [80] and EpiTEome [81] include the analysis of split reads that cross junctions between TEs and uniquely mappable genome regions.

BSMAP [76], published in 2009, is a general-purpose bisulfite reads mapping algorithm for the analysis of whole-genome shotgun BS-seq data. BSMAP addressed the issues faced when mapping high-throughput bisulfite reads to the reference genome. Those issues include increased searching space, reduced complexity of bisulfite sequence, asymmetric cytosine to thymine alignments, and multiple CpG heterogeneous methylation. The algorithm aligns with a wildcard approach: it enumerates all possible combinations of C/T conversion in the BS read to find the uniquely mapping position with the least mismatches on the reference genome and it supports gapped/pair-end alignment, iterative trimming of low-quality base pairs, and multi-thread parallel computing.

BS Seeker [82] converts the genome to a three-letter alphabet and aligns the bisulfite-treated reads to a reference genome using the Bowtie aligner [83]. The algorithm is able to work with data from both BS-seq [75] and MethylC-seq [84] protocols. Mapping ambiguity is reduced by accounting for tags that are generated by certain library construction protocols [75]. Post-processing of the alignments removes non-unique and low-quality mappings based on the (user-defined) allowed number of mismatches. The performance of the method was compared with the previously published methods RMAP [85], Maq [86], and BSMAP [76] on simulated 36-mer BS reads from human chromosome 21 that were mapped using both BS-seq and MethylC-seq protocols. The results showed that when the protocol used includes tags [75], BS Seeker has the highest accuracy and the shortest run time (around 4 min for 1 million reads); otherwise its performance is very comparable to RMAP.

Two updated versions followed in 2013 and 2018, respectively: BS-Seeker2 [87] and BS-seeker3 [79]. The former improved the mappability by using local alignment and it provides additional filtering out of reads with incomplete BS conversion, while the latter offers ultra-fast aligning of the reads and better visualization of the methylation data.

Bismark [77] was the first published BS-Seq aligner to handle single- and paired-end mapping of both directional and non-directional bisulfite libraries. The methylation output discriminates between sequence context (CpG, CHG, or CHH, where H is any base but G) and can be also obtained in an alignment strand-specific format, a very useful option to study asymmetric methylation (hemi- or CHH methylation) in a strand-specific manner. Bismark, like BS Seeker, adopts an “in silico bisulfite conversion” strategy, where all the Cs in both the reads and the reference are converted to Ts prior to alignment, thus resulting in a three-letter genome. A direct comparison of Bismark with BS Seeker [82] returned a very similar number of alignments in a similar time scale and efficiency.

MOABS [78] is a pipeline written in C++ that detects differential methylation with 10-fold coverage at single-CpG resolution based on a Beta-Binomial hierarchical model and is capable of processing two billion reads in 24 CPU hours. The method captures both sampling and biological variations, it corrects the depth bias that arises from the fact that sequencing depth is normally higher (or lower) in high (or low) copy-number regions, and it provides a single innovative metric that reports on the combined biological and statistical significance of differential methylation. MOABS integrates a number of BS-seq procedures, e.g., read mapping, methylation ratio calling, identification of hypo- or hyper- methylated regions from one sample, and differential methylation from multiple samples.

MOABS outperforms other algorithms, such as Fisher’s exact test [74] and BSmooth (the first program that accounted for biological variation using a modified t-test) [88] on simulated and real BS-seq data, especially at low sequencing depth. MOABS also offers the possibility to extend the analysis to differential 5-hydroxymethylcytosine (5hmC) analysis using the Reduced Representation Bisulfite Sequencing (RRBS) [89] protocol and Oxidative Bisulfite Sequencing (oxBS-seq) [90].

In *Arabidopsis thaliana*, DNA methylation occurs in three DNA sequence contexts: mCG, mCHG, and mCHH. A great source of genetic differences between individuals is the variation in TE content [91], which highlights the importance of the identification of TE variants. In 2016, Stuart et al. introduced TEPID (Transposable Element Polymorphism IDentification) [80], an algorithm for accurately mapping the locations of TE presence/absence variants with respect to a reference genome. The analysis of TE methylation levels is reinforced by including the analysis of split reads that cross junctions between TEs and uniquely mappable genome regions. The authors applied the method on genome resequencing data for 216 different Arabidopsis accessions [92]. The analysis discovered, among other findings, that the majority of the TE variants were due to the de novo insertion of TEs, while a smaller subset was possibly due to the deletion of ancestral TE copies, mostly around the pericentromeric regions. Furthermore, it was found that a TE insertion is connected with an increase in flanking DNA methylation levels, whereas the deletion of an ancestral TE was often not associated with a corresponding decrease in flanking DNA methylation levels.

The methylation analysis of non-reference and mobile TEs required both genome resequencing and MethylC-seq datasets. EpiTEome [81] is the first pipeline that combines the detection of new TE insertion sites, and the methylation states of the insertion and the surrounding site from a single MethylC-seq dataset. The preprocessing step includes trimming and filtering the reads and mapping them to the reference genome using Bismark [77] or any other MethylC-seq mapping program. EpiTEome functions similarly to TEPID [80] to identify new TE insertion sites. The novelty of epiTEome is the ability to detect the DNA methylation status of the transposed element and the insertion site using the exact split-reads that identified the transposition event. EpiTEome reports the DNA methylation as single-insertion alignments, as well as a meta-analysis of all insertion sites in a sample.

Bicycle [93] is a pipeline designed to analyze DNA methylation data from BS sequencing experiments. Comparison with an array of publicly available pipelines demonstrates that Bicycle provides a number of features that most other tools lack, thus making it the most complete bioinformatic pipeline for the analysis of bisulfite sequencing data. The supplementary material of the above publication [93] provides a thorough comparison of Bicycle with other pipelines used for the analysis of bisulfite sequencing data.

A number of additional improvements for methylation analysis have been proposed. Among them, a method to assess the low mappability of young TEs, like L1-Ta, in the human genome, was repurposed by Shukla et al. [94] so as to align BS reads to a consensus sequence. Noshay et al. [95] described an interesting method that first rigorously determines the average genome-wide bisulfite conversion rates, and subsequently uses this as a parameter to deal with mapping ambiguities from differences in conversion rates. DNA methylation analysis, however, still remains a challenging bioinformatic task that requires further study.

### 2.2. Analysis of Small RNA Expression

Small RNAs (sRNAs) can act either transcriptionally by leading the epigenetic modifications at TE loci, or post-transcriptionally through targeted RNA degradation. sRNAs of the PIWI-interacting RNA (piRNA) class are the most potent silencers of TEs in germline cells [24]. In somatic tissues, two additional classes of small RNAs contribute to TE silencing: short interfering RNAs (siRNAs) derived from expressed transposon transcripts [81] and the more recently described 30 tRNA-derived fragments (30 tRFs) [96]. Thus, it becomes evident that considering sRNAs and accurately quantifying their production is pivotal to the study of transposon biology. To this end, several packages have been released to investigate sRNA classes, which prove particularly challenging when derived from repetitive loci in the genome as they are short in length, typically between 18–36 nucleotides. Some methods have largely considered sRNA classes separately; however, several packages (e.g., unitas [97] and TEsmall [98]) consider sRNA classes comprehensively to facilitate proper normalization of heterogeneous sRNA libraries, and to facilitate differential expression analysis across classes while taking into consideration ambiguously mapped reads.

Even though microRNAs (miRNAs) are not considered to be largely involved in the regulation of TEs, a non-negligible fraction of miRNAs appears in a large number of copies in the genome [99,100,101], thus highlighting the significance of tools for the study of TEs that provide multimapped reads. For more on the topic of miRNAs and TEs, there is a number of relevant publications [102,103,104,105,106].

#### 2.2.1. Tools for the Analysis of Multiple sRNA Types

In 2008, Moxon et al. presented a toolkit for analyzing large-scale plant small RNA datasets [107], especially micro RNAs and trans-acting siRNAs (ta-siRNAs) that can both induce post-transcriptional silencing of target genes. The web-based tools can identify mature miRNAs and their precursors, compare sRNA expression profiles under varying conditions or between mutants and wild-type, and predict ta-siRNA. The successor of the toolkit is the UEA sRNA workbench [108], a downloadable suite of tools that provides a user-friendly platform to create workflows for processing sRNA next-generation sequencing data. The workbench offers an enhanced version of the functions of its predecessor, as well as complemented with easily accessible complementary visualization tools.

MiRanalyzer [109] is a web server tool that aimed to perform the analysis of the upcoming large amount of sRNAs deep-sequencing data. Using a list of unique miRNA reads and their expression levels, MiRanalyzer detects the known miRNAs, maps the remaining reads against transcribed sequences, and predicts new microRNAs. The tool is based on a random forest learning scheme that employs a selection of features (associated with nucleotide sequence, structure, and energy) based on their information gain. The prediction model was built on datasets from Human, rat, and *C. elegans* and reaches AUC values of 97.9% and recall values of up to 75% on unseen data.

SeqBuster [110] is another web-based tool that analyzes large-scale sRNA datasets, and the first to characterize isomiRs [111], miRNA variants, that usually arise as a result of enzymatic 5′- or 3′-trimming, 3′ nucleotide addition or nucleotide substitution) [112,113,114,115]. The packages perform a variety of analyses, including identification of sRNAs, distribution of their length and frequency, and comparative expression levels of different sRNA loci between different samples. Application of SeqBuster to small-RNA datasets of human embryonic stem cells revealed that most miRNAs present different types of isomiRs, some of them being associated with stem cell differentiation. The authors also provide a stand-alone version, which allows for annotation against any custom database.

DARIO [116] is a web service that allows the study of short read data from sRNA-seq experiments. Using mapped reads as an input, DARIO performs quality control, overlaps them with user-selected gene models, quantifies the RNA expression based on annotated ncRNAs from different ncRNA databases, and predicts new ncRNAs via a random forest classifier. DARIO supports the following assemblies as reference genomes: human (hg18 and hg19), mouse (mm9 and mm10), Rhesus monkey, Zebrafish, *C. elegans*, and *D. melanogaster*.

miRDeep2 [117] is a user-friendly update and extension of miRDeep [118], an algorithm that uses a probabilistic model of miRNA biogenesis to score the fit of sequenced RNAs to the biological model of miRNA biogenesis. The improved algorithm identifies canonical and non-canonical miRNAs such as those derived from TEs and informs on high-confidence candidates that are detected in at least two independent samples. miRDeep2 was tested on high-throughput sequencing data from seven animal species representing the major animal clades. In all clades tested, the algorithm identified miRNAs with high accuracy (98.6–99.9%) and sensitivity (71–90%), and it reported hundreds of novel miRNAs.

miRTools [119] is a web service for the classification of sRNAs, annotation of known miRNAs based on NGS data, prediction of novel miRNAs, and identification of differentially expressed miRNAs. A few years later, its improved version, mirTools 2.0 [120] offered a series of additional features: detection and profiling of more types of ncRNAs (such as tRNAs, snRNAs, snoRNAs, rRNAs, and piRNAs), identification of miRNA-targeted genes, detection of differentially expressed ncRNA, as well as a standalone version of the tools. However, the webserver of mirTools is currently inaccessible.

ShortStack [121] is a stand-alone application that allows for the analysis of reference-aligned sRNA-seq data and de novo annotation and quantification of the inferred sRNA genes. It provides highly specific annotation of miRNA loci in all tested plant (*Arabidopsis*, tomato, rice, and maize) and animal (*D. melanogaster*, mouse, and human) species. ShortStack reports on parameters relevant to sRNAgene annotation, such as size distributions, repetitiveness, strandedness, hairpin-association, miRNA annotation, and phasing. ShortStack uses modest computational resources and has comparable performance with previously published tools (e.g., UEA sRNA workbench) upon testing on sRNA-seq data set from wild-type *Arabidopsis* leaves.

sRNAtoolbox [122] is a web-interfaced set of interconnected tools, including expression profiling from deep sequencing data via the sRNAbench tool [123]), consensus differential expression, consensus target prediction, blast search against several remote databases, and visualization of sRNAs (differential) expression. All tools can be used independently or for the exploration and downstream analysis of sRNAbench results. An updated version of the sRNAtoolbox [124] features additions such as new reference genomes from Ensembl, bacteria and virus collections from NCBI, and microRNA reference sequences from MirGeneDB, as well as parallel launching of several jobs (batch mode).

Chimira [125] is a web-based system for fast analysis of miRNAs from small RNA-seq data and identification of epi-transcriptomic modifications (5′- and 3′-modifications, internal modifications, or variation), based on which it can identify global microRNA modification profiles. The input sequences are automatically cleaned, trimmed, size selected, and mapped directly to miRNA hairpin sequences. Chimira offers a set of tools for the interpretation and visualization of the results that facilitates the comparative analysis of the input samples. The results from benchmarking show that Chimira offers faster execution than Oasis [126].

The accurate annotation and analysis of short non-coding RNAs (sncRNAs) often required the installation of multiple tools with possibly different technical limitations (e.g., dependencies, operating system). Gebert et al. [97] developed unitas, a software that provides complete annotation in a manner suitable for non-expert users. By using a single tool, one can overcome the issue of the normalization of multiple mapping sequences. unitas supports the species with available ncRNA reference sequences in the Ensembl databases and provides standalone precompiled for Linux, Mac, and Windows systems.

Published in 2018, TEsmall [98] is a novel software package that allows for simultaneous mapping, annotation, and relative quantification of a variety of sRNAs types including structural RNAs, miRNAs, siRNAs, and piRNAs on a common scale. Thus, it enables the study of the expression trends among different sRNA types and provides an insight into the cross-talk between sRNA regulatory pathways. Given the appropriate annotation, TEsmall can provide the same functions for any novel type of sRNA. TEsmall can shed light on the complex regulatory networks of different types of sRNAs that act cooperatively, especially in the area of transposon silencing. It is known that siRNAs and piRNAs repress TEs in somatic cells and the germline, respectively, but also piRNAs are found to act in conjunction with siRNAs to perform this role, whereas in plants miRNAs might serve as an intermediate to form siRNAs [127,128,129].

Oasis 2 [130] is a web application useful for detecting and classifying sRNAs, as well as for analyzing their differential expression. Oasis 2 is a faster and more accurate version of Oasis [126] (accuracy of around 87% for Oasis 2 versus 80% for the original application) that also recognizes potential cross-species miRNAs and viral and bacterial sRNAs in infected samples and provides the option for interactively visualizing novel miRNAs and querying them against 14 supported genomes or the Oasis database of miRNAs and miRNA families.

sRNAPipe [131] is a user-friendly pipeline that offers a range of analyses for small RNA-seq data. The pipeline performs successive steps of mapping small RNA-seq reads to chromosomes, TEs, gene transcripts, miRNAs, small nuclear RNAs, rRNAs, and tRNAs. It also provides individual mapping, counting, and normalization for chromosomes, TEs, and gene transcripts, and tests ping-pong amplification for putative piRNAs. sRNAPipe allows for the rapid and precise analysis of high-throughput data and it generates publication-quality figures and graphs. It is available in both the Galaxy Toolshed and via GitHub.

Also based on the Galaxy framework, RNA workbench 2.0 [132] is a comprehensive set of analysis tools and consolidated workflows. It integrates an abundance (more than 100) tools useful in the field of RNA research, such as RNA alignment, annotation, target prediction, and RNA-RNA interaction.

GeneTEflow [133] is a fully automated, reproducible, and platform-independent workflow that allows the comprehensive analysis of both genes and locus-specific TEs expression from RNA-Seq data employing different technologies (Nextflow [134] and Docker [135]). The pipeline can be extended to include additional types of analysis such as alternative splicing and fusion genes.

Manatee [136] is an algorithm for the quantification of sRNA classes. In contrast to many available sRNA analysis pipelines, Manatee rescues highly multimapping and unaligned reads based on available annotation and robust density information and is capable of identifying and quantifying expression from isomiRs and unannotated loci that could give rise to yet unknown sRNAs. Performance comparison on real and simulated data shows that other state-of-the-art methods (among them ShortStack [121] and sRNAbench [124]) tend to overestimate transcripts with zero abundance in the simulated dataset and underestimate/assign zero reads to expressed and highly expressed transcripts. On the other hand, Manatee estimates counts that are the closest to the simulated abundances and achieves high accuracy across diverse sRNA classes. Moreover, Manatee can be easily implemented in pipelines, and its output is suitable for downstream analyses and functional studies.

Di Bella et al. [137] published an elaborate comparative analysis of eight pipelines on RNA-seq data, including Oasis 2, sRNApipe, and sRNA workbench. Their systematic performance evaluation aims at establishing guidelines for the selection of the most appropriate workflow for each ncRNA class.

#### 2.2.2. Tools for the Analysis of PIWI-Interacting RNAs

PIWI-interacting RNAs (or piRNAs) are animal-specific RNAs that comprise the largest and most heterogeneous class of the small ncRNA (sncRNA) family, with over 2 million distinct piRNA species in mouse [138]. They function as guides for PIWI proteins, a subfamily of Argonaute proteins. Their length is 21–35 nucleotides and they are processed from long single-stranded precursor transcripts that originate from genomic loci known as piRNA clusters. piRNA clusters have been found to contain remnants of transposons in arthropods, whereas in birds and mammals they encode for long non-coding RNAs that are processed into piRNAs. In the majority of mammalian species, some RNAs are involved in the protection of the germline genome against transposon mobilization. piRNAs are mostly not conserved among species [139]. PIWI-interacting RNAs (piRNAs) were first identified as novel silencing RNAs in the *Drosophila melanogaster* testis two decades ago [140].

Active retrotransposition is more frequent in germ cells due to the epigenetic reprogramming that primes them for totipotency [141]. To maintain the integrity of the genome passed on to the next generation, the metazoan germline exhibits the so-called piRNA pathway, an additional retrotransposon control based on small RNA-mediated recognition and endonucleolytic cleavage of the target TE transcripts [142,143]. piRNAs are loaded into PIWI proteins and thus target the TE transcripts by sequence complementarity. The TE transcripts are subsequently cleaved, producing secondary piRNAs, which constitutes the so-called “ping-pong” cycle in fruit fly [59,144,145,146,147]. The presence and functions of piRNAs in somatic cells are not as well characterized; however, it is known that some piRNAs are common for the germline and the soma, some appear exclusively in the soma, whereas others are exclusive for each tissue type [148]. piRNAs were found to exhibit a bias for starting with a “U” in the 1st position and an “A” at the 10th position [149].

The first bioinformatics tool for piRNA prediction was a *k*-mer scheme (2011) [150] which applied the Fisher discriminant algorithm to *k*-mer (*k* = 1 through 5) sequence features using small RNA data. The study trained the algorithm on datasets from non-piRNA and piRNA sequences of five model species (rat, mouse, human, fruit fly, and nematode), and it reports precision and sensitivity of over 90% and over 60%, respectively. The authors conclude that the method can be used to identify piRNAs of non-model organisms without complete genome sequences; however, the web server is currently out of order.

Pibomd (2014) [151] is an SVM algorithm for piRNA identification based on motif discovery. Pibomd employed the computational biology tool Teiresias [152] to identify motifs of variable length that appear frequently in mouse piRNA and non-piRNA sequences and developed an SVM classifier that uses those motifs as features. Training of an imbalanced SVM classifier (Asym-Pibomd) on the same training and testing datasets provided higher specificity but lower sensitivity than the balanced SVM classifier; however still higher sensitivity and accuracy than the *k*-mer scheme [150] on identifying mouse piRNAs. Analysis of the distribution of the motifs showed uniform distribution of motifs in the non-piRNA sequences but significant motif enrichment on the 5′- and/or 3′-end of the piRNA sequences. The web server allows users to upload multiple FASTA sequences and select the model for classification (balanced or imbalanced SVM classifier). The performance of the algorithm on datasets from five model species (rat, mouse, human, fruit fly, and nematode) is comparable to the *k*-mer scheme [150] (see Table 1).

Genomic alignment of small RNA-seq data is a critical methodology for the study of small RNA. Butter (2014) [153] is a Perl wrapper for samtools and for the short read aligner bowtie (ultrafast, memory-efficient short read aligner geared toward quickly aligning large sets of short DNA sequences to large genomes) [83,154] to produce small RNA-seq alignments where multimapped small RNAs tend to be placed near regions of confidently high density.

PIANO (2014) [155] is a program for piRNA annotation that uses piRNA-transposon interaction information predicted by RNAplex [156]. piRNAs are aligned to transposons and a support vector machine (SVM) is applied on triplet elements that combine structure and sequence information extracted from piRNA-transposon matching/pairing duplexes. The SVM classifier can predict human, mouse, and rat piRNAs, with overall accuracy greater than 90%.

Luo et al. introduced a method for differentiating transposon-derived piRNAs (2016) [157] based on six sequence-derived features. The datasets were derived from NONCODE version 3.0 for 3 species, namely human, mouse, and *Drosophila* with a 1:1 ratio of positive to negative samples so that the results could be compared with those from previous studies [155]. The study adopted two approaches: direct combination, which merges different feature vectors, and ensemble learning, which uses the weighted average scores of individual feature-based predictors; however, the weights are determined arbitrarily based on the AUC scores of the base predictors. The prediction models employ a random forest as the main classifier engine 10-fold cross. Validation of both methods on the human dataset achieved AUC and accuracy of at least 90% and 80% respectively on all three datasets. Both methods yield higher AUC scores upon comparison with the *k*-mer method [150] and PIANO [155].

Another study by the same researchers developed a genetic algorithm-based weighted ensemble method named GA-WE (2016) [158] for predicting transposon-derived piRNAs was trained on piRNA datasets from human, mouse, and *D. melanogaster*, shown in Table 2. The GA-WE models, in contrast with the previously described work [157], determine automatically the optimal weights on the validation set. The method achieves AUC of at least 0.93 on both the balanced and unbalanced datasets by 10-fold cross-validation and it produces lower scores for cross-species experiments, indicating that piRNAs derived from different species may have different patterns. By adopting their previous work [157], PIANO [155], and the *k*-mer scheme [150] as benchmark methods, 10-CV on the human dataset showed that GA-WE achieves higher AUC scores on all datasets. Regarding cross-species prediction, models constructed on the mouse dataset perform better on the human dataset, possibly because of similar piRNA length distribution between the two mammals but different length distribution between mouse and *Drosophila*.

piPipes (2015) [159] is a set of five pipelines for the analysis of piRNA/transposon from different Next Generation Sequencing libraries (small RNA-seq, RNA-seq, Genome-seq, ChIP-seq, CAGE/Degradome-Seq). piPipes allows for the analysis of a single library and pair-wise comparison between two samples. It is implemented in Bash, C++, Python, Perl, and R and provides a standardized set of tools to analyze these diverse data types.

Liu et al. (2017) introduced a two-layer ensemble classifier, 2L-piRNA [160], that first addressed the double question: can we predict a piRNA based solely on sequence information, and can we distinguish whether it is of the type that instructs DNA deadenylation? In its first layer, 2L-piRNA identifies whether a query RNA molecule is a piRNA or not, while in the second layer it identifies whether a piRNA has (or not) the function of instructing target mRNA deadenylation. The authors constructed a benchmark dataset consisting of 709 piRNA sequences that have the function of instructing target mRNA deadenylation, 709 piRNA sequences that do not have that function (extracted from piRBase), and 1418 non-piRNA sequences. The sequences were represented using pseudo-K-tuple nucleotide composition (PseKNC) [161] for K = 2, and six helical parameters (rise, roll, shift, slide, tilt, and twist) for each possible RNA dinucleotide were taken into account. Comparison of the performance of the method with the “Accurate piRNA prediction” by Luo et al. [157] and GA-WE [158] showed that 2L-piRNA outperforms them on all metrics, as seen in Table 3.

In 2014, Brayet et al. proposed piRPred [162], an extensible and adaptive classification method for piRNA prediction that combines heterogeneous types of piRNA features and allows for the implementation of newly discovered piRNA characteristics. piRPred fuses support vector machines (SVMs) and multiple kernels that represent the following features: frequency of certain *k*-mer motifs, the presence of a Uridine base as the first nucleotide of the sequence, the distance to centromeres and telomeres, and the occurrence of piRNAs in clusters in the genome. The algorithm was trained on human and *Drosophila* piRNAs (positive datasets) and tRNAs, mature miRNAs, and exonic sequences (negative dataset). Comparison with the *k*-mer method (see Table 4) proposed by Zhang et al. [150] shows that piPRed performs better, especially on human piRNA data.

In 2015, Menor et al. [163] introduced a method for piRNA and mature miRNA classification based on the previously described Multiclass Relevance Units Machine classifier (McRUM) [164], an empirical Bayesian kernel method. piRNA datasets were extracted from NONCODE 3.0 and the sequences were represented using *k*-mers, for *k* = 1 through 5. The authors made use of correlation-based feature selection (CFS) to select a subset of features on which to build classifier models and reduce the dimensionality. Comparison with the *k*-mer scheme [150], both the original and the one retrained on the datasets of the McRUM-based approach, reveals that the sensitivity of the latter is roughly 60% higher.

V-ELMpiRNApred (2017) [165] is based on an ensemble classifier called voting-based extreme learning machine (V-ELM). It implements a hybrid feature vector of *k*-mer features (*k* = 1 through 5) and short sequence motifs (SSM), a series of new features with 80 dimensions that allow the study of the relation between discontinuous sites of sequences. Feature selection is then used to remove the *k*-mer features with redundant information. V-ELMpiRNApred was trained on human piRNA and non-piRNA sequences from NONCODE 3.0 and its classification performance was compared with methods published earlier. Table 5 shows that V-ELMpiRNApred outperforms previously published methods on all metrics.

In the same year, Boucheham et al. introduced IpiRId [166], which allows for the representation of different types of features by combining several kernels that can be tested independently, thus enabling the study of feature conservation across species. The features include genomic and epigenomic information; apart from the sequence, IpiRId takes into consideration the positions on the chromatin, the positions regarding the sequence and/or structural motifs that can occur at the 5′ and/or the 3′ ends, possible occurrence in clusters, and interaction with specific target sequences. IpiRId, at its core, is based on the Multiple Kernel Learning (MKL), which allows for combining heterogeneous features by automatically tuning their weights in order to improve the prediction. Comparison of the performance with previously published techniques on datasets consisting of piRNAs from human, mouse, and *Drosophila* shows that IpiRId outperforms the rest of the techniques scoring more than 90% accuracy in all species and similar values in all the other metrics (see Table 6). To be noted is that Piano was originally trained on piRNA datasets from *Drosophila*. Moreover, the study of the pertinence of the features best represented across species reveals that the most conserved piRNA features are Uridine and Adenine in the first and tenth position respectively, occurrence in clusters, and binding with transposons.

In 2018, Wang et al. introduced piRNN [167], the first deep learning program for piRNA identification, which is based on convolutional neural networks (CNN) and adopts a genome-independent approach that does not need genome and/or epigenomic data for identifying piRNAs. piRNN constructs a feature vector from the input sequences in two parts: first, it extracts the *k*-mer (*k* = 1, 2, 3, 4, 5) motif frequencies and second it updates the feature vector with *k*-mers motifs around the 1st and 10th base if the sequence starts with a T/U and/or has an A in the 10th position. Comparison of the performance with Piano, 2L-piRNA, and the *k*-mer scheme on human data (as representative of mammalian piRNAs) and *D. melanogaster* piRNA data shows that piRNN outperforms the other methods on all metrics used (see Table 7).

The authors provide four models trained on piRNA data for four species (human, rat, *C. elegans,* and *D. melanogaster*); if desired, the users can retrain the models or train new ones. All in all, piRNN is a useful tool for piRNA prediction in non-model organisms with limited genomic resources.

piRNAPred [168] (2019) is an integrated framework for piRNA prediction that employs hybrid features like *k*-mer nucleotide composition (k-KNC, k = 1 to 5, which is a sort of k-mer strings “normalized” for the sequence length), secondary structure (paired or unpaired state), and thermodynamic and physicochemical properties of contiguous dinucleotides, extracted from piRNA sequences of eight species. Comparison of the performance of the best performing piRNAPred model with previously published methods reveals that the former exhibits the highest accuracy and a Matthew’s Correlation Coefficient (MCC) of 0.97 (Table 8).

#### 2.2.3. Tools for the Analysis of Circular RNAs

Thirty years ago, circular RNAs (circRNAs) were described as “abnormally spliced transcripts” formed by scrambled exons [169], a phenomenon known as “exon shuffling” or “non-co-linear splicing”. circRNAs produced by co- and posttranscriptional head-to-tail “backsplicing”, where an exon’s 3′ splice site is ligated onto an upstream 5′ splice site of an exon on the same RNA molecule, as well as circRNAs generated from intronic lariats during colinear splicing, may exhibit physiologically relevant regulatory functions in eukaryotes [170]. It has been demonstrated that circRNA production and canonical pre-mRNA splicing compete with each other and some splicing factors like *muscleblind* might interact with flanking introns to promote exon circularization [171]. CircRNAs are abundant in eukaryotic cells; measurements in human fibroblast cells revealed that there are over 25,000 circRNA isoforms per cell [172]. Up to 23% of the actively transcribed human genes give rise to circRNAs whose expression is dynamically regulated between tissues, cell types, and during differentiation [173]. Genome-wide studies revealed that half of the circRNAs do not contain intervening introns, whereas in hematopoietic progenitor cells introns are retained in 20% of the circRNAs [174]. Pseudogenes can be retrotranscribed from circRNAs and they can also be inherited in mammalian genomes [175,176]. Bioinformatic analysis has shown that the intronic flanks adjacent to circularized exons are enriched in complementary ALU repeats [172], with ALUs being the most abundant TEs [177]. The alternative formation of inverted repeated Alu pairs and the competition between them can mediate alternative circularization, which leads to multiple circRNA transcripts [178]. A recent study [176] has indicated that circRNAs and TEs possibly co-evolve in a species-specific and dynamic manner. Their findings suggest a model according to which many circRNAs emerged convergently during evolution, as a byproduct of splicing in orthologs prone to insertion of TEs.

DeepCirCode [179] utilizes a 2-layer convolutional neural network (CNN) to predict back-splicing for the formation of human circRNAs. The model takes as an input the binary vector (one-hot encoding) of the intron and exon sequences flanking the potential back-splicing sites and it predicts whether the two sites can be back-spliced. The kernels in the first layer detect the motif sites related to back-splicing, whereas the kernels in the second layer learn more complex motifs. The model was trained on human exonic circRNAs from the publicly accessible databases circRNADb [180] and circBase [181]. The performance of the model was compared with an SVM and an RF model [182] (see Table 9), previously constructed by the authors, that use *k*-mer compositional features.

Relevant features learned by DeepCirCode are represented as sequence motifs, some of which match human known motifs involved in RNA splicing, transcription, or translation. Analysis of these motifs shows that their distribution in RNA sequences can be important for back-splicing. Moreover, some of the human motifs appear to be conserved in mouse and fruit fly.

Previously published bioinformatics and Machine Learning methods are suitable for animal circRNAs. Plants are rich in splicing signals and transposable elements, and the characteristics of their circRNAs are different from those in animals. Yin et al. [183] recently presented PCirc, a method for extracting a variety of features (including open reading frames, numbers of *k*-mers, and splicing junction sequence coding) from rice circRNAs and trained a machine learning model based on a random forest algorithm. The classification of PCirc was evaluated by accuracy, precision, and F1 score, all of which scored above 0.99 when using rice circRNAs and lncRNAs as positive and negative datasets respectively. Testing the model on other plant datasets yielded accuracy scores larger than 0.8.

A summary of the methods discussed above, including the year of publication and the web address in which the code/data (if any) are deposited, is provided in Table 10.

## 3. Conclusions

TE are abundant across various organisms, and understanding the diverse roles and the mechanisms of their regulation is a challenging and complex task. Advancements in sequencing techniques provide better annotated reference genomes, and next-generation sequencing offers ultra-high-throughput data. Even though several tools have been developed for the analysis of sequencing data, individual packages are not straightforward nor easy to use for non-expert users. However, a number of pipelines have been published that combine a set of tools in an efficient and user-friendly manner. Another challenge for the analysis tools is the ambiguity of the reads that are produced by bisulfite sequencing. Many tools deal with the issue by excluding those multimapping reads from their analysis, whereas others employ various methods to reduce the mapping ambiguity.

At the same time, even though high-performing specialized tools for the analysis of a specific sRNA type have been developed, methods that provide simultaneous mapping and annotation of several types of sRNAs can shed light on the cross-talk between sRNA regulatory pathways and uncover sRNAs that act in conjunction.

To sum up, the development of bioinformatics and machine learning methods for the analysis of the sequencing data can offer valuable qualitative and/or quantitative insight into the mechanisms that regulate the expression of TEs. These methods include a variety of individual tools, pipelines, and ML algorithms that focus on the major regulatory mechanisms: DNA methylation and several types of small non-coding RNAs that act separately or cooperatively for the silencing of TEs.

## Figures and Tables

**Table 1 biology-10-00896-t001:** Comparison of the performance of the imbalance and balanced SVM classifier of Pibomd with the *k*-mer scheme.

Method	Sp (%)	Sn (%)	Acc (%)
*k*-mer scheme [136]	98.4	52.04	75.22
Pibomd	89.76	91.48	90.62
Asym-Pibomd	96.2	72.68	84.44

**Table 2 biology-10-00896-t002:** The datasets from three model organisms used for training the ensemble method.

Species	Raw Real Pirnas	Raw Non-Pirna Ncrnas	No. of Transposons
Human	32.152	59.003	4679.772
Mouse	75.814	43.855	3660.356
*Drosophila*	12.903	102.655	37.326

**Table 3 biology-10-00896-t003:** 2L-piRNA outperforms the previously published methods across all the metrics used.

Method	Sn (%)	Sp (%)	Acc (%)	MCC
1st layer				
2L-piRNA	88.3	83.9	86.1	0.723
Accurate piRNA prediction [157]	83.1	82.1	82.6	0.651
GA-WE [158]	90.6	78.3	84.4	0.694
2nd layer				
2L-piRNA	79.1	76.0	77.6	0.552
Accurate piRNA prediction	N/A	N/A	N/A	N/A
GA-WE	N/A	N/A	N/A	N/A

**Table 4 biology-10-00896-t004:** piRPred outperforms the k-mer scheme, especially upon testing on the human piRNA dataset.

Method	Human			*Drosophila*		
	Sn (%)	Sp (%)	Acc (%)	Sn (%)	Sp (%)	Acc (%)
piRPred	0.88	0.84	0.86	0.83	0.95	0.89
*k*-mer scheme [150]	0.30	0.82	0.58	0.45	0.92	0.69

**Table 5 biology-10-00896-t005:** Comparison of the performance of V-ELMpiRNApred with earlier methods.

Method	Sn (%)	Sp (%)	Acc (%)	MCC (%)
V-ELMpiRNAPred	95.6	94.8	95.2	0.899
piRPred [162]	82.5	88.3	85.4	0.709
*k*-mer scheme [150]	86.7	52.1	69.4	0.414
Asym-Pibomd [151]	92.7	91.3	92.0	0.840
Piano [155]	93.8	91.6	92.7	0.854
McRUMs [164]	93.9	92.3	93.1	0.862

**Table 6 biology-10-00896-t006:** Comparison of the performance of IpiRId with earlier methods.

Method/ Species	Human				Mouse				*Drosophila*			
	Sn (%)	Sp (%)	Acc (%)	Pre (%)	Sn (%)	Sp (%)	Acc (%)	Pre (%)	Sn (%)	Sp (%)	Acc (%)	Pre (%)
*k*-mer scheme [150]	48.40	95.5	71.85	91.49	47.79	94.10	70.95	89.01	63.90	40.45	52.17	51.76
Piano [155]	0	100	50	0	0	100	50	0	78.90	96.90	87.90	96.22
Pibomd [151]	78.05	78.21	78.13	78.17	79.43	78.82	79.13	78.94	70.44	61.72	66.08	64.78
piRPred [162]	80.54	81.86	81.2	81.67	90.36	91.48	90.92	91.39	86	86.72	86.36	86.66
IpiRId	90.56	89.62	90.09	89.73	90.74	96.58	93.66	96.37	87.27	97.90	92.59	97.67

**Table 7 biology-10-00896-t007:** piRNN outperforms the selected methods on piRNA datasets from both model organisms.

Method/ Species	Human				*Drosophila*			
	Sn	Sp	Acc	Pre	Sn	Sp	Acc	Pre
*k*-mer scheme [150]	0.55	0.89	0.72	0.84	0.14	0.93	0.53	0.66
Piano [155]	0.92	0.32	0.62	0.58	0.87	0.50	0.68	0.63
2L-piRNA [160]	0.79	0.51	0.67	0.68	0.39	0.71	0.52	0.65
piRNN	0.97	0.97	0.95	0.94	0.97	0.97	0.95	0.93

**Table 8 biology-10-00896-t008:** Summary of the performance of piRNAPred and previously published methods on piRNA datasets from 8 species.

Model	Species	Sn (%)	Sp (%)	Acc (%)	MCC
*k*-mer scheme [150]	Homo sapiens, Mus musculus, Drosophila melanogaster, Caenorhabditis elegans	72.47	95.5	NA	NA
Piano [155]	Drosophila melanogaster	95.89	94.6	95.27	NA
Pibomd [151]	Mus musculus	91.48	89.8	90.62	NA
Accurate piRNA prediction [157]	Homo sapiens, Mus musculus, Drosophila melanogaster	83.10	82.10	82.6	0.651
GA-WE [158]		90.6	78.3	84.4	0.694
2L-piRNA [160]	Mus musculus	88.3	83.9	86.1	0.723
piRNApred	Homo sapiens, Mus musculus, Drosophila melanogaster, Caenorhabditis elegans, Danio *rerio*, Gallus gallus domesticus, Xenopus tropicalus, Bombyx mori	98.57	98.6	98.6	0.97

**Table 9 biology-10-00896-t009:** Performance comparison of DeepCircCode with two models developed previously by the authors.

Model	Sn (%)	Sp (%)	Acc (%)	MCC
DeepCirCode	92.14	76.84	85.24	70.38
SVM [182]	86.23	68.29	77.23	55.40
RF [182]	83.82	74.36	79.07	58.41

**Table 10 biology-10-00896-t010:** Bioinformatics and ML tools for exploring the regulation of TEs and the repository for the relevant data/code.

	Method	Year	Code Availability
DNA methylation analysis			
1	Maq [86]	2008	http://maq.sourceforge.net/ (accessed on 29 June 2021)
2	RMAP [85]	2008	http://rulai.cshl.edu/rmap/ (accessed on 29 June 2021)
3	BSMAP [76]	2009	https://code.google.com/archive/p/bsmap/ (accessed on 29 June 2021)
4	BS Seeker [82]	2010	http://pellegrini-legacy.mcdb.ucla.edu/bs_seeker/BS_Seeker.html (accessed on 29 June 2021)
5	Bismark [77]	2011	https://github.com/FelixKrueger/Bismark (accessed on 29 June 2021)
6	BSmooth [88]	2012	http://rafalab.jhsph.edu/bsmooth (accessed on 29 June 2021)
7	BS-Seeker2 [87]	2013	http://pellegrini-legacy.mcdb.ucla.edu/bs_seeker2/ (accessed on 29 June 2021)
8	MOABS [78]	2014	https://code.google.com/archive/p/moabs/ (accessed on 29 June 2021)
9	TEPID [80]	2016	https://github.com/ListerLab/TEPID (accessed on 29 June 2021)
10	EpiTEome [81]	2017	https://github.com/jdaron/epiTEome (accessed on 29 June 2021)
11	BS-Seeker3 [69]	2018	https://github.com/khuang28jhu/bs3 (accessed on 29 June 2021)
12	Bicycle [93]	2018	http://www.sing-group.org/bicycle/ (accessed on 29 June 2021)
13	GeneTEFlow [133]	2020	https://github.com/zhongw2/GeneTEFlow (accessed on 29 June 2021)
sRNAs analysis			
14	UEA sRNA toolkit [107]	2008	http://srna-tools.cmp.uea.ac.uk (accessed on 29 June 2021)
15	miRDeep [118]	2008	
16	miRanalyzer [109]	2009	https://bioinfo2.ugr.es/ceUGR/miranalyzer/ (accessed on 29 June 2021)
17	mirTools 1.0 [119]	2010	centre.bioinformatics.zj.cn/mirtools/ (accessed on 29 June 2021)
18	SeqBuster [110]	2010	https://github.com/lpantano/seqbuster (accessed on 29 June 2021)
19	DARIO [116]	2011	http://dario.bioinf.uni-leipzig.de/index.py (accessed on 29 June 2021)
20	miRDeep2 [117]	2012	https://github.com/rajewsky-lab/mirdeep2 (accessed on 29 June 2021)
21	UEA sRNA workbench [108]	2012	http://srna-workbench.cmp.uea.ac.uk (accessed on 29 June 2021)
22	mirTools 2.0 [120]	2013	http://122.228.158.106/mr2_dev (accessed on 29 June 2021)
23	ShortStack [121]	2013	https://github.com/MikeAxtell/ShortStack (accessed on 29 June 2021)
24	sRNAbench [114]	2014	https://arn.ugr.es/srnatoolbox/srnabench/ (accessed on 29 June 2021)
25	Chimira [125]	2015	http://wwwdev.ebi.ac.uk/enright-dev/chimira/ (accessed on 29 June 2021)
26	sRNAtoolbox [122]	2015	https://arn.ugr.es/srnatoolbox/ (accessed on 29 June 2021)
27	Oasis [126]	2015	http://oasis.dzne.de (accessed on 29 June 2021)
28	Unitas [97]	2017	https://sourceforge.net/projects/unitas/ (accessed on 29 June 2021)
29	TEsmall [98]	2018	https://github.com/mhammell-laboratory/TEsmall (accessed on 29 June 2021)
30	Oasis 2 [130]	2018	https://oasis.dzne.de/ (accessed on 29 June 2021)
31	sRNAPipe [131]	2018	https://github.com/GReD-Clermont/sRNAPipe (accessed on 29 June 2021)
32	RNA workbench 2.0 [132]	2019	https://github.com/bgruening/galaxy-rna-workbench (accessed on 29 June 2021)
33	Manatee [136]	2020	https://github.com/jehandzlik/Manatee (accessed on 29 June 2021)
piRNAs analysis			
34	*k*-mer scheme [150]	2011	
35	Butter [153]	2014	https://github.com/MikeAxtell/butter (accessed on 29 June 2021)
36	PIANO [155]	2014	http://www.insect-genome.com/links/piano.php (accessed on 29 June 2021)
37	Pibomd [151]	2014	http://app.aporc.org/Pibomd/ (accessed on 29 June 2021)
38	piRPred [162]	2014	https://github.com/IshaMonga/piRNAPred (accessed on 29 June 2021)
39	piPipes [159]	2015	https://github.com/bowhan/piPipes (accessed on 29 June 2021)
40	McRUM-based [164]	2015	not found
41	Ensemble learning [157]	2016	
42	GA-WE [158]	2016	https://github.com/zw9977129/piRNAPredictor (accessed on 29 June 2021)
43	2L-piRNA [160]	2017	bioinformatics.hitsz.edu.cn/2L-piRNA/ (accessed on 29 June 2021)
44	V-ELMpiRNAPred [165]	2017	http://mm20132014.wicp.net:38601/velmprepiRNA/Main.jsp (accessed on 29 June 2021)
45	IpiRId [166]	2017	https://evryrna.ibisc.univ-evry.fr/evryrna/ (accessed on 29 June 2021)
46	piRNN [167]	2018	https://github.com/bioinfolabmu/piRNN (accessed on 29 June 2021)
47	piRNApred [165]	2019	https://github.com/IshaMonga/piRNAPred (accessed on 29 June 2021)
circRNAs analysis			
48	DeepCirCode [179]	2019	https://github.com/BioDataLearning/DeepCirCode (accessed on 29 June 2021)
49	PCirc [183]	2021	https://github.com/Lilab-SNNU/Pcirc (accessed on 29 June 2021)

## Data Availability

Not applicable.

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
