# Peer review of "Bioinformatics and Machine Learning Approaches to Understand the Regulation of Mobile Genetic Elements"

_biology, 2021, doi:10.3390/biology10090896_

Round 1
Reviewer 1 Report
The review paper “Bioinformatics and Machine Learning Approaches to Understand the Regulation of Mobile Genetic Elements” has summarized the recently available tools to understand the transposable elements (TEs) regulation and function. More specifically, authors summarized the tool used for analysis of DNA methylation and small RNA types that includes the machine learning approaches. In general, TEs are well-known phenomenon, yet it has not been explored much in terms of TE regulation and function. There are many research articles and reviews about understand TE regulation and function. This review article summarized the recent tools and methodology which will be helpful for the researchers working on TEs and related activities. Overall, the article well written and focused in-depth and summarized relevant and recent information in a good possible way. Furthermore, considering the scope of this journal Biology, I can recommend the editor may consider this manuscript for publication, but I have few concerns to be addressed.
My major concerns of this manuscript are lack of in-depth information in some aspects.
In section 2.1 Analysis of DNA methylation
In this section author has given elaborate description on most of the DNA methylation methods however it lacks comparative description with other methods. It is not clear which method is best on what aspects - . Perhaps make a figure or table to give a better comparative overview on the DNA methylation analysis methods.
Similarly, a better comparative overview of the tool described in section “2.2.1. Tools for the analysis of multiple sRNA types” that might be helpful to the readers.
Authors completely omitted the long-read based DNA methylome analysis which will be important for overcoming some of the shoet-read DNA methylome analysis that mentioned in the conclusion such as sequence ambiguity and multimapping (line 605-607).
References need to be checked again some references missing doi information (ex. reference 40).
The manuscript is well written but overall it requires a proof editing a little more.
Author Response
Dear Reviewer,
Thank you for the comments - please find attached a document with point by point reply and edits made to the manuscript.

Reviewer 2 Report
ID: biology-1300698
Type of manuscript: Review
Title: Bioinformatics and Machine Learning Approaches to Understand the Regulation of Mobile Genetic Elements Special Issue: Regulation of Mobile Genetic Elements at the Molecular Level
This Review is very timely and needed. In a way, its value may be compared to that of Dan L. Lindsley, E. H. Grell. 1967, Genetic Variations of Drosophila Melanogaster and Maniatis T., Sambrook J., Fritsch E.F. Molecular Cloning: A Laboratory Manual. Indeed, this review is a laboratory manual in Bioinformatics, as stated by authors:
A summary of the methods discussed above, including the year of publication and |
593 |
the web address in which the code/data (if any) are deposited, is provided in Table 10. |
594 |
595 |
|
Table 10. Bioinformatics and ML tools for exploring the regulation of TEs and the repository for the |
596 |
relevant data/code |
|
It should be published with minor corrections: the sentences in Introduction are too long, it is better to split one phrase in two or three. Seemingly, this may be done by the editors or by authors. Also, “Mappability“ is rather an unusual word.
Author Response

(The authors gave the same response as above.)

Reviewer 3 Report
The manuscript titled "Bioinformatics and Machine Learning Approaches to Understand the Regulation of Mobile Genetic Elements" by Giassa and Alexiou aims to review key analytical aspects of the transposable elements (TE) and their regulations. While the manuscript is well organized and easy to read, it digresses at several places and discusses general computational tools without any specific reference to the TE. My specific comments are given below.
(1) While discussing DNA-methylation, authors discuss about tools like BSMAP, BS seeker Bismark, MOAB and Bicycle without discussing about their utility in TE analysis. Authors can easily point to other reviews that cover these tools in more detail and focus and discuss in this review on studies that utilized these tools in exploring TE-DNAmethylation.
(2) GeneTEflow is an RNA-Seq based tools. However it is included in the section of DNA-methylation.
(3) More introduction is required on small RNA and piRNA sections to link it with TEs. Both these sections describe the tools in great details. However, these sections fail to first establish how piRNA, sRNAs and CircRNA regulate TEs and their expression.
(4) Page 6, line 246-249- this statement should have a reference.
Author Response

(The authors gave the same response as above.)

Round 2
Reviewer 3 Report
Authors have sufficiently addressed my previous concerns.
Author Response
We thank the reviewer for positive reply to our comments.